# TAILORING LINEAR MODELS
# FOR JOINT REPRESENTATION

## ABSTRACT

*Piecewise Linear Approximation* (PLA) calls to represent a long data series by line segments under a maximum error threshold. State-of-the-art PLA methods save space over *lossless compression* by quantizing starting points and jointly representing similar segments, yet they neither tailor segments for joint representation nor minimize segment count. In this paper, we present TAILORPIECE, a suite of lossy PLA algorithms that explicitly tailor PLA segments for short sequence length and joint representation under an error threshold and starting-value quantization. TAILORPIECEDP optimizes a *mergeability* criterion; in a degenerate form, it reduces to minimum-length PLA. TAILORPIECEGD greedily selects segment endpoints within a tunable search space, enabling farther extension of subsequent segments and balancing compression with runtime. Experiments show TAILORPIECEDP raises compression ratio by up to $34\%$, while TAILORPIECEGD gains similar savings with two orders of magnitude lower runtime.

## 1 INTRODUCTION

Sectors like healthcare, food supply, and transportation increasingly rely on high-frequency time-series data from diverse sources (Botta et al., 2016; Xu et al., 2014; Atzori et al., 2010), to support automation, monitoring, and other advances (Gupta et al., 2020). Yet the sheer data volume renders storage costly (Jensen et al., 2018). Various encodings advance *lossless* floating-point compression (Liakos et al., 2022; 2024; Kuschewski et al., 2023; Afroozeh et al., 2023), offering gains over the widely used lossless compression algorithm, Gorilla (Pelkonen et al., 2015). Still, even these representations usually attain a compression ratio below 2, hence remain costly (Chen et al., 2024).

*Lossy* compression offers an alternative to lossless methods for storing large time-series datasets, allowing control of space requirements via a tunable *maximum error threshold*. Modern Time-Series Management Systems (TSMS) (Jensen et al., 2018; 2019) let users find the shortest Piecewise Linear Approximation (PLA) sequence that approximates a time-series within a desired maximum error threshold (Elmeleegy et al., 2009; Hakimi & Schmeichel, 1991) to meet their compression needs. The $L_\infty$ norm target is often preferred to $L_1$ or $L_2$,

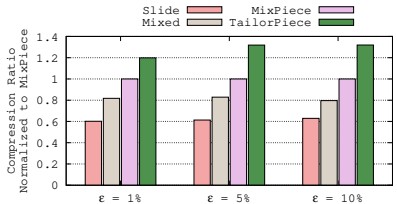

Figure 1: TAILORPIECE enhances PLA sequence length and mergeability.

as it bounds the error for each data record rather than just in aggregate (Karras & Mamoulis, 2008; Luo et al., 2015). A recent proposal, MIXPIECE (Kitsios et al., 2024), compresses time-series with maximum error guarantees by quantizing PLA segment starting values by the given error threshold and *jointly* representing segments having common starting values and overlapping allowable *slopes*, yielding extra space savings as Figure 1 shows. However, MIXPIECE does not ensure minimum sequence length (i.e., number of segments) nor configures segments for joint representation. Methods that minimize sequence length under an error bound (Elmeleegy et al., 2009; Hakimi & Schmeichel, 1991) disregard *quantized* starting values and are therefore inapplicable.

In this paper, we propose TAILORPIECE, a suite of algorithms that tailor PLA segments aiming at small sequence length and joint representation:

- MINSEGMENTS, a dynamic programming algorithm that returns a *minimum-length* PLA representation with quantized starting values under a maximum error threshold.

- TAILORPIECEDP, which, building on top of MINSEGMENTS, produces segments with wide permissible slope intervals to enhance their mergeability, and merges them.
- TAILORPIECEGD, a greedy algorithm that selects each segment's end to maximize the *next* segment's reach and allows tuning its endpoint search space to trade runtime for compression.

Our algorithms unlock the potential of grouping short PLA sequences, improving the average compression ratio by up to 34% over MIXPIECE, as Figure 1 shows. Remarkably, TAILORPIECEGD achieves slightly larger space savings than MINSEGMENTS, as it produces PLA segments more likely to be grouped, while being two orders of magnitude faster.

## 2 BACKGROUND AND RELATED WORK

**PLA with maximum error guarantees**  PLA represents a series of timestamped values $\langle t_i, v_i \rangle_{i \geq 1}$ by *line segments*. Some PLA methods *join* consecutive segments at their *knots* (Elmeleegy et al., 2009; Gritzali & Papakonstantinou, 1983; Hakimi & Schmeichel, 1991), others assume *disjoint* knots (Stone, 1961; Pavlidis, 1973; O'Rourke, 1981; Elmeleegy et al., 2009), and some consider both (Luo et al., 2015). We consider *disjoint* knots, where each segment may be non-continuous with its predecessor. We describe each segment by its start timestamp $t_i$, value $v_i$, and slope $a_i$. Common norms are $L_2$ (Euclidean distance) and $L_\infty$ (maximum absolute error). We focus on $L_\infty$, to keep *each* value within error $\epsilon$. SLIDE (Elmeleegy et al., 2009; O'Rourke, 1981) finds the *minimum-length* disjoint PLA sequence under a maximum error threshold, greedily building the convex hull of data points in the segment under construction to maintain the admissible slope range.

---

**Algorithm 2.1:** MIXPIECE$_{phase1}$

**Input:** A data signal $s$: $\langle t_i, v_i \rangle \ \forall i \in \{1, \ldots, n\}$, and an error threshold $\epsilon$
**Output:** An associative array $b\_intervals$, mapping each quantized $b$ value to a list of tuples $\langle a_l, a_u, t \rangle$

```
1  Function MIXPIECE phase1(s, ε)
2      b_intervals ← {{}, . . . , {}};
3      ⟨ts, vs⟩ ← s.next();
4      b⁻ ← ⌊vs/ε⌋ε; b⁺ ← ⌈vs/ε⌉ε; // quantized starting points
5      a_u⁻ ← ∞; a_l⁻ ← −∞; a_u⁺ ← ∞; a_l⁺ ← −∞; // slope intervals
6      floor ← true; ceil ← true; diff ← 0;
7      while s.hasNext() do
8          ⟨tc, vc⟩ ← s.next();
9          if vc > a_u⁻(tc − ts) + b⁻ + ε or vc < a_l⁻(tc − ts) + b⁻ − ε then
10             floor ← false; // stop b⁻ segment
11         if vc > a_u⁺(tc − ts) + b⁺ + ε or vc < a_l⁺(tc − ts) + b⁺ − ε then
12             ceil ← false; // stop b⁺ segment
13         if floor then diff + +;
14         if ceil then diff − −;
15         if !floor and !ceil then // if both segments stopped use larger
16             if diff > 0 then b_intervals[b⁻].add(⟨a_l⁻, a_u⁻, ts⟩);
17             else b_intervals[b⁺].add(⟨a_l⁺, a_u⁺, ts⟩);
18             ⟨ts, vs⟩ ← ⟨tc, vc⟩;
19             b⁻ ← ⌊vs/ε⌋ε; b⁺ ← ⌈vs/ε⌉ε;
20             a_u⁻ ← ∞; a_l⁻ ← −∞; a_u⁺ ← ∞; a_l⁺ ← −∞;
21             floor ← true; ceil ← true; diff ← 0;
22         if vc < a_u⁻(tc − ts) + b⁻ − ε then // lower slope
23             a_u⁻ ← (vc+ε−b⁻)/(tc−ts);
24         if vc > a_l⁻(tc − ts) + b⁻ + ε then // raise slope
25             a_l⁻ ← (vc−ε−b⁻)/(tc−ts);
26         if vc < a_u⁺(tc − ts) + b⁺ − ε then // lower slope
27             a_u⁺ ← (vc+ε−b⁺)/(tc−ts);
28         if vc > a_l⁺(tc − ts) + b⁺ + ε then // raise slope
29             a_l⁺ ← (vc−ε−b⁺)/(tc−ts);
30     if diff > 0 then b_intervals[b⁻].add(⟨a_l⁻, a_u⁻, ts⟩);
31     else b_intervals[b⁺].add(⟨a_l⁺, a_u⁺, ts⟩);
32     return b_intervals;
```

---

**MIXPIECE**  (Kitsios et al., 2024), the leading PLA method, *quantizes* segment starting values and *jointly* represents segments with common starting values and overlapping admissible *slope intervals* with a *minimum* number of groups by partitioning an *interval graph*, whose edges denote overlapping intervals, into the fewest *cliques* (Kitsios et al., 2023) in $O(n \log n)$ time (Gupta et al., 1982).

Algorithm 2.1 outlines the first phase of MIXPIECE. The input comprises a sequence of discrete data points, $\langle t_i, v_i \rangle$, $i \in \{1, \ldots, n\}$, and an error threshold $\epsilon$. Line 4 forms the key difference from other PLA algorithms (Elmeleegy et al., 2009), as MIXPIECE quantizes an original value $v$ to $b^-$ and $b^+$, i.e., the nearest lower and higher multiples of $\epsilon$:

$$b^- = \lfloor v/\epsilon \rfloor \times \epsilon$$
$$b^+ = \lceil v/\epsilon \rceil \times \epsilon \tag{1}$$

For instance, with $\epsilon = 0.5$, both original values 1.1 and 1.4 yield $b^- = 1$ and $b^+ = 1.5$. In this way, MIXPIECE derives a limited amount of *discrete* starting values shared among line segments, facilitating *joint* representation.

The remainder of Algorithm 2.1 performs angle-based PLA, maintaining two upper and and lower slopes (Line 5) that form an angle anchored at the two running starting points, $b^-$ and $b^+$, and containing each subsequent signal point. Each of the two pairs of slopes defines the segment's *slope interval*. When the distance of an encountered point deviates by more than $\epsilon$ from the angle formed by both pairs of bounding slopes, $a_{u-}$ and $a_{l-}$, and $a_{u+}$ and $a_{l+}$ (Line 12), the greedy algorithm terminates the two running segments and adds the longest one to the list of intervals, with the respective starting point, $b^-$ or $b^+$, attributed with the respective final bounding slopes and the starting point's timestamp $t_s$ (Lines 16 and 17), and reiterates with the newly encountered point $\langle t_c, v_c \rangle$ as starting point for the next segment (Line 18). Otherwise, MIXPIECE may lower the upper slopes $a_{u-}$ and $a_{u+}$ (Lines 23 and 27) or raise the lower slopes $a_{l-}$ and $a_{l+}$ (Lines 25 and 29) to accommodate the new point. Algorithm 2.1 processes each data point in $O(1)$, hence its time complexity is $O(n)$.

## 3 OVERVIEW

We aim to reduce PLA storage under a maximum-error threshold $\epsilon$. While MIXPIECE (Kitsios et al., 2024) merges segments to minimize groups, it does *not* minimize PLA segments for a given quantization and $\epsilon$. We first solve this problem with MINSEGMENTS, a dynamic programming algorithm. Inspired therefrom, TAILORPIECEDP enhances segment mergeability for extra space savings, and TAILORPIECEGD attains similar savings at two orders of magnitude lower runtime.

### 3.1 QUANTIZED REACH

As a preparatory step, we extract Procedure 3.1 from MIXPIECE (Kitsios et al., 2024), which selects the longest among the linear segments starting from each original value $v$, $\epsilon$-*quantized* to the nearest lower $b^-$ or higher $b^+$ multiple of $\epsilon$, with $b^- = \lfloor v/\epsilon \rfloor \cdot \epsilon$, $b^+ = \lceil v/\epsilon \rceil \cdot \epsilon$. For instance, with $\epsilon = 0.5$, each of values 1.1 and 1.4 yields $b^- = 1$ and $b^+ = 1.5$. Figure 2 illustrates the process. Two angles, one initiated from $\langle t_1, b^- \rangle$ (Figure 2a) with bounding slopes $a_{u_2}^-$ and $a_{l_2}^-$ (Line 13– Line 14) and one from $\langle t_1, b^+ \rangle$ (Figure 2b) with bounding slopes $a_{u_2}^+$

**Procedure 3.1:** $\epsilon\_quantized\_reach(s, i, \epsilon)$

**input** : Starting index $i$, data signal $s$: $\langle t_i, v_i \rangle \, \forall i \in \{1, \ldots, n\}$, error threshold $\epsilon$
**output** : $\epsilon$-quantized reach of $i$
1 $reach \leftarrow 0$; $s.seek(i)$; $\langle t_s, v_s \rangle \leftarrow s.next()$;
// quantize $v_s$ to nearest lower ($b^-$) and higher ($b^+$) multiples of $\epsilon$
2 $b^- \leftarrow \lfloor v_s/\epsilon \rfloor \epsilon$; $b^+ \leftarrow \lceil v_s/\epsilon \rceil \epsilon$;
3 $a_u \leftarrow \infty$; $a_l \leftarrow -\infty$; $a_{u+} \leftarrow \infty$; $a_{l+} \leftarrow -\infty$;
4 $floor \leftarrow true$; $ceil \leftarrow true$; $reach \leftarrow 0$;
5 **while** $s.hasNext()$ **do**
6 $\quad \langle t_c, v_c \rangle \leftarrow s.next()$;
7 $\quad$ **if** $v_c > a_{u-}(t_c - t_s) + b^- + \epsilon$ **or** $v_c < a_{l-}(t_c - t_s) + b^- - \epsilon$ **then**
8 $\quad\quad floor \leftarrow false$; // stop $b^-$ segment
9 $\quad$ **if** $v_c > a_{u+}(t_c - t_s) + b^+ + \epsilon$ **or** $v_c < a_{l+}(t_c - t_s) + b^+ - \epsilon$ **then**
10 $\quad\quad ceil \leftarrow false$; // stop $b^+$ segment
11 $\quad$ **if** $floor$ **or** $ceil$ **then** $reach++$; // within bounds
12 $\quad$ **else return** $reach$; // out of bounds
13 $\quad$ **if** $v_c < a_{u-}(t_c - t_s) + b^- - \epsilon$ **then** $a_{u-} \leftarrow \frac{v_c + \epsilon - b^-}{t_c - t_s}$; // lower slope
14 $\quad$ **if** $v_c > a_{l-}(t_c - t_s) + b^- + \epsilon$ **then** $a_{l-} \leftarrow \frac{v_c - \epsilon - b^-}{t_c - t_s}$; // raise slope
15 $\quad$ **if** $v_c < a_{u+}(t_c - t_s) + b^+ - \epsilon$ **then** $a_{u+} \leftarrow \frac{v_c + \epsilon - b^+}{t_c - t_s}$; // lower slope
16 $\quad$ **if** $v_c > a_{l+}(t_c - t_s) + b^+ + \epsilon$ **then** $a_{l+} \leftarrow \frac{v_c - \epsilon - b^+}{t_c - t_s}$; // raise slope
17 **return** $reach$;

and $a_{l_2}^+$ (Line 15–Line 16), both subtended by $\langle t_2, v_2 + \epsilon \rangle$ and $\langle t_2, v_2 - \epsilon \rangle$, enclose all lines starting from $b^-$ or $b^+$ that approximate the two seen points within $\epsilon$. As the next point, $\langle t_3, v_3 \rangle$, lies within both angles, yet more than $\epsilon$ away from their upper and lower slopes, we reduce them to $(a_{u_3}^-, a_{l_3}^-)$ (Figure 2a) and $(a_{u_3}^+, a_{l_3}^+)$ (Figure 2b), subtended by $\langle t_3, v_3 + \epsilon \rangle$ and $\langle t_3, v_3 - \epsilon \rangle$. Next, point $\langle t_4, v_4 \rangle$ lies outside the angle formed by $(a_{u_3}^-, a_{l_3}^-)$ (Figure 2a), hence cannot be approximated by a segment starting at $b^-$ (Line 10). Contrariwise, a further reduction of the angle starting from $b^+$ approximates

point $\langle t_4, v_4 \rangle$ with $\epsilon$, we thus set its upper slope to $a_{u_4}^+$, connecting $\langle t_1, b^+ \rangle$ to $\langle t_4, v_4 + \epsilon \rangle$ and retain the lower slope $a_{l_3}^+$, already within $\epsilon$ of $\langle t_4, v_4 \rangle$. The gray area in Figure 2b captures the candidate lines within $\epsilon$ of all four points.

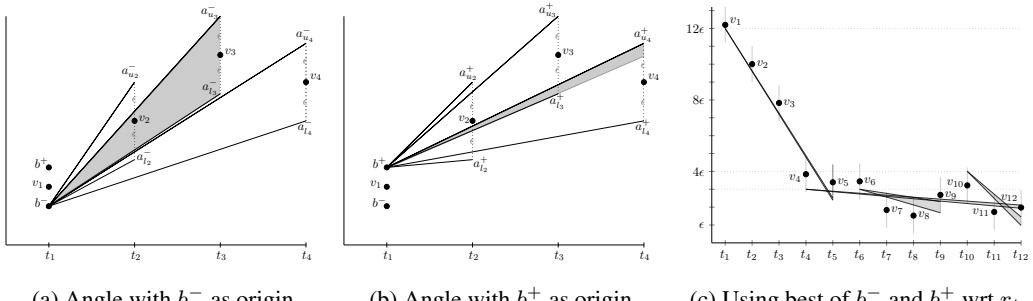

(a) Angle with $b^-$ as origin.      (b) Angle with $b^+$ as origin.      (c) Using best of $b^-$ and $b^+$ wrt $r_i$.

Figure 2: MIXPIECE opts for the segment starting from $b^+$ (Figure 2b) and reaching $t_4$ over that starting from $b^-$ (Figure 2a) reaching $t_3$. Figure 2c shows the longest segments from $t_1, t_4, t_6, t_{10}$.

**Definition 1.** *The $\epsilon$-quantized reach $r_i$ of timestamp $t_i$ in $s = \langle t_i, v_i \rangle$ is the maximum length of a linear segment starting from an $\epsilon$-quantized $v_i$ that approximates subsequent values within $\epsilon$.*

**Lemma 1.** *Procedure 3.1 returns the $\epsilon$-quantized reach of $i$.*

*Proof.* Assume Procedure 3.1 returns $r_i$ for $\langle v_1, t_1 \rangle$, while a segment of length $r_i + 1$ starting from a quantization of $v_1$ stays within $\epsilon$ up to $\langle t_{i+1}, v_{i+1} \rangle$. By design, Procedure 3.1 tracks this line up to $t_i$, hence cannot have stopped at $t_i$. By contradiction, Procedure 3.1 finds the maximum length. $\qquad\square$

We configure Procedure 3.1 to return the longest segment's *bounding slopes*, along with its length.

### 3.2 MINIMIZING THE NUMBER OF SEGMENTS

The greedy strategy of Procedure 3.1, used in (Kitsios et al., 2024), maximizes *reach* from a given point but ignores global optimality. On the signal of Figure 2c, it selects segment $[t_1, t_5]$, then $[t_6, t_9]$, and $[t_{10}, t_{12}]$. Yet, as Figure 3 shows, reach $r_4$ spans farther than $r_6$ and $r_{10}$ combined, so two segments—$[t_1, t_3]$ and $[t_4, t_{12}]$—suffice to approximate the signal. This counterintuitive outcome arises be-

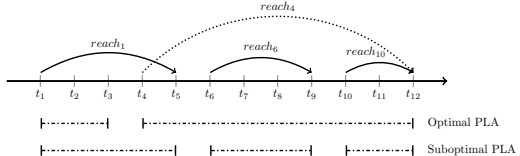

Figure 3: Maximally extending each segment may lead to a suboptimal sequence length: $r_4$ surpasses $r_6 + r_{10}$, hence we reach $t_{12}$ with two segments by ending the first segment at $t_3$.

cause $r_i$, being dependent on the quantization of $v_i$ to either $b_i^-$ or $b_i^+$, exceeds $\delta i + r_{i+\delta i}$, hence starting at $i$ is preferable to starting at $i + \delta i$. The greedy algorithm is thus suboptimal. We define the problem of finding a minimum-length PLA under quantization as follows:

**Problem 1.** *Given data sequence $s: (t_i, v_i), i \in \{1, \ldots, n\}$ and error threshold $\epsilon$, find a minimum-length PLA sequence of disjoint segments from $\epsilon$-quantized values, to approximate $s$ within $\epsilon$.*

MINSEGMENTS (Algorithm 3.2) solves Problem 1 by dynamic programming, getting each starting point's *reach* (Line 3) via Procedure 3.1 and recursively deriving the least PLA length from $t_i$ (Line 8):

$$
L(i) = \begin{cases} \min_{i < j \leq i + r_i} \{L(j+1) + 1\}, & i < n \\ 1, & i = n \\ 0, & i > n \end{cases} \quad (2)
$$

**Algorithm 3.2:** MINSEGMENTS $(s, \epsilon)$

> **input** : Signal $s$: $(t_i, v_i) \, \forall i \in \{1, \ldots, n\}$, error threshold $\epsilon$
> **output** : A minimum-length PLA sequence on $s$
> 1   $i \leftarrow 1$;
> 2   **while** *s.hasNext()* **do**
> 3     $r[i] \leftarrow \epsilon$.quantized_reach(s, i, $\epsilon$);
> 4     s.next();
> 5     $i$++;
> 6   $i \leftarrow N$;
> 7   **while** $i \geq 1$ **do**
> 8     Compute $L[i]$; // by Equation (2)
> 9     $i$ − −;
> 10   **return** $L[1]$;

**Theorem 1.** MINSEGMENTS *solves Problem 1 optimally.*

*Proof.* The proof follows by induction. By the inductive hypothesis, for all $j > i$, $L(j)$ gives the minimum PLA length for the subsequence starting at $t_j$. Each valid endpoint $j$ of the first segment, $i < j \leq i + r_i$, yields $L(j+1) + 1$ segments in total. Minimizing over all such $j$ therefore gives the minimum PLA length starting from $t_i$. $\qquad\square$

Figure 4 depicts MINSEGMENTS's computation for the signal of Figure 2c. The left side shows the *reach* of starting points, while the right side computes the optimal PLA sequence length starting from each point via Equation (2). Since $r_1 = 4$, the first segment may end at any of $\langle t_2, t_3, t_4, t_5 \rangle$. With $L(2)$, $L(3)$, and $L(5)$ larger than 1, and $L(4) = 1$, we end the first segment at $t_3$ and approximate the signal using only *two* segments, exploiting the large reach $r_4$. MINSEGMENTS (Algorithm 3.2) returns a globally optimal solution to Problem 1 via the recursive minimization in

$$reach_{11} = 1 \quad L(11) = 1$$
$$reach_{10} = 2 \quad L(10) = 1$$
$$reach_9 = 3 \quad L(9) = 1$$
$$reach_8 = 4 \quad L(8) = 1$$
$$reach_7 = 5 \quad L(7) = 1$$
$$reach_6 = 3 \quad L(6) = 1 + L(10) = 2$$
$$reach_5 = 4 \quad L(5) = 1 + L(10) = 2$$
$$reach_4 = 8 \quad L(4) = 1$$
$$reach_3 = 2 \quad L(3) = 1 + L(6) = 3$$
$$reach_2 = 2 \quad L(2) = 1 + L(4) = 2$$
$$reach_1 = 4 \quad L(1) = 1 + L(4) = 2$$

$$t_1 \; t_2 \; t_3 \; t_4 \; t_5 \; t_6 \; t_7 \; t_8 \; t_9 \; t_{10} \; t_{11} \; t_{12}$$

Figure 4: Computing least PLA sequence length.

Equation (2) in $O(Rn)$ time, where $R$ is the maximum reach in the signal, as each recursion step is linear in $r_i$ for each $i$. For small maximum error thresholds, $R$ is typically a small constant. Our implementation of Algorithm 3.2 also returns the starting points and *bounding slopes* (as in Section 3.1) of segments in the minimum-length PLA sequence, along with the sequence length.

### 3.3 TAILORPIECEDP ALGORITHM

Each segment in the PLA of Algorithm 3.2 has two *bounding slopes* defining its admissible slope range for line segments that approximate the data therein. We define *slope interval size* as follows.

**Definition 2.** *The* slope interval size $I_k$ *of segment* $k^{\text{th}}$ *is the gap between its upper and lower slopes.*

In Figure 2a, $I_1 = a_{u_3}^- - a_{l_3}^-$, and in Figure 2b, $I_1 = a_{u_4}^+ - a_{l_3}^+$.

By MIXPIECE's storage model, we aim to merge and jointly represent segments with coinciding starting points and overlapping slope intervals. However, Equation (2) may miss the best compression: it shortens the PLA sequence yet overlooks the slope interval sizes of the segments it creates. *Larger* intervals are preferable, as they are more likely to overlap. Accordingly, we enhance MINSEGMENTS to TAILORPIECEDP, which yields segments with larger slope intervals to boost overlap among them and enable further segment grouping. To craft TAILORPIECEDP, we refine TAILORPIECEDP's objective to a composite function $C(i)$ that favors both large slope intervals and few segments starting at timestamp $t_i, i \geq 1$. The average slope interval size over $L(i)$ segments is $\frac{\sum_{k=1}^{L(i)} I_k}{L(i)}$. To modulate the influence of individual slope interval sizes, we introduce an exponent $p \in [0, 1]$ on the numerator terms, yielding $\frac{\sum_{k=1}^{L(i)} I_k{}^p}{L(i)}$. To ease the recursion, we define the numerator aggregate as $S(i) = \sum_{k=1}^{L(i)} I_k{}^p$. For $p = 0$, the fraction equals 1, as the number of segments divided by itself. For $p > 0$, it favors many short segments that yield large slope intervals. To counter this effect and favor fewer segments, we additionally normalize by $L(i)$, yielding a squared denominator in our composite objective: $C(i) = \frac{S(i)}{L(i)^2}$.

**Problem 2.** *Given a sequence* $s$: $(t_i, v_i)$ *and error threshold* $\epsilon$, *find a PLA of disjoint segments starting from* $\epsilon$-quantized *values that approximates* $s$ *within* $\epsilon$ *and maximizes* $C(i) = \frac{S(i)}{L(i)^2}$.

Equation (3) solves Problem 2 by recursively maximizing $C(i)$; $I(i, j)$ is the slope interval size of segment $[t_i, t_j]$, while $S(i)$ and $L(i)$ track the optimizing numerator and denominator at each step.

$$C(i) = \begin{cases} \max_{i < j \leq i + r_i} \left\{ \frac{S(j+1) + I(i,j)^p}{(L(j+1)+1)^2} \right\} & i < n \\ 1 & i = n \\ 0 & i > n \end{cases} \tag{3}$$

TAILORPIECEDP replaces $L[i]$ with $\{C[i], S[i], L[i]\}$ in Lines 8 and 10 and returns $C[1]$ in place of $L[1]$ in Line 10 of Algorithm 3.2. Exponent $p$ enables fine-grained control and links TAILORPIECEDP to MINSEGMENTS: $p = 0$ reduces TAILORPIECEDP to minimizing segments, as MINSEGMENTS does. In our experiments, TAILORPIECEDP with a broad range of $p$ values outperform MINSEGMENTS across datasets.

### 3.4 TAILORPIECEGD ALGORITHM

TAILORPIECEDP seeks a PLA sequence that approximates a signal within a threshold $\epsilon$ using segments with $\epsilon$-quantized starting values and maximizes $C(i) = \frac{S(i)}{L(i)^2}$. However, optimality comes at the cost of higher computational overhead.

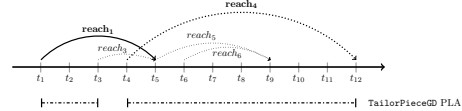

Figure 5: TAILORPIECEGD ends the first segment at $t_3$ to reach $t_{12}$ with two segments.

We hence propose TAILORPIECEGD, a greedy algorithm balancing the efficiency of MIX-PIECE (Kitsios et al., 2024) with the effectiveness of TAILORPIECEDP. It reduces MIX-PIECE' myopic behavior by enhancing lookahead when forming a segment. Figure 5 illustrates an case where a segment starting at $t_1$ may reach $t_5$. While MIXPIECE would create this segment and begin the next one at $t_6$, TAILOR-PIECEGD may end the first segment earlier—at $t_2, t_3$ or $t_4$—to allow a longer subsequent reach. Ending at $t_3$ is locally optimal, as the long reach of $r_4$ extends the next segment to $t_{12}$, while other endpoints may only reach $t_9$. TAILORPIECEGD, outlined in Algorithm 3.3, considers several candidate endpoints $j$ beyond the default $i + r_i$ used by MIXPIECE. For each $j$, it considers the reach of the next segment starting at $j + 1$, and selects the *earliest* point $j_{\min}$ among those maximizing the *combined reach* of the two segments. Choosing the *earliest* favors larger slope intervals, as the breadth of a slope interval is non-increasing with segment length, and thereby enables better grouping. The exhaustive search of endpoints shortens PLA sequences and broadens slope intervals, yet incurs

---

**Algorithm 3.3:** TAILORPIECEGD $(s, \epsilon)$

**Input:** Data signal $s$: $\langle t_i, v_i \rangle \ \forall i \in \{1, \ldots, n\}$, error threshold $\epsilon$
**Output:** Array $b\_intervals$ mapping each quantized value to $\langle a_l, a_u, t \rangle$ tuple list

```
1  Function TAILORPIECEGD (s, ε)
2      b_intervals ← {{}, ..., {}}; ⟨t_s, v_s⟩ ← s.next();
3      b⁻ ← ⌊v_s/ε⌋ε; b⁺ ← ⌈v_s/ε⌉ε; // quantized starting points
4      a_u- ← +∞; a_l- ← -∞; a_u+ ← +∞; a_l+ ← -∞; // slope intervals
5      floor ← true; ceil ← true; diff ← 0; i ← 1;
6      j_min ← min{arg max_{j∈[i+r_i^q,i+r_i]}{j + r_{j+1}}}; // Equation (4)
7      while s.hasNext() do
8          ⟨t_c, v_c⟩ ← s.next();
9          i ++;
10         if v_c > a_u-(t_c - t_s) + b⁻ + ε  or  v_c < a_l-(t_c - t_s) + b⁻ - ε then
11             floor ← false;
12         if v_c > a_u+(t_c - t_s) + b⁺ + ε  or  v_c < a_l+(t_c - t_s) + b⁺ - ε then
13             ceil ← false;
14         if floor then diff ++;
15         if ceil then diff --;
16         if i > j_min then // close segment when reaching j_min
17             if diff > 0 then b_intervals[b⁻].add(⟨a_l-, a_u-, t_s⟩);
18             else b_intervals[b⁺].add(⟨a_l+, a_u+, t_s⟩);
19             ⟨t_s, v_s⟩ ← ⟨t_c, v_c⟩;
20             b⁻ ← ⌊v_s/ε⌋ε; b⁺ ← ⌈v_s/ε⌉ε;
21             a_u- ← +∞; a_l- ← -∞; a_u+ ← +∞; a_l+ ← -∞;
22             floor ← true; ceil ← true; diff ← 0;
23             j_min ← min{arg max_{j∈[i+r_i^q,i+r_i]}{j + r_{j+1}}};
                   // Eq. (4)
24         if v_c < a_u-(t_c - t_s) + b⁻ - ε then // lower slope
25             a_u- ← (v_c+ε-b⁻)/(t_c-t_s);
26         if v_c > a_l-(t_c - t_s) + b⁻ + ε then // raise slope
27             a_l- ← (v_c-ε-b⁻)/(t_c-t_s);
28         if v_c < a_u+(t_c - t_s) + b⁺ - ε then // lower slope
29             a_u+ ← (v_c+ε-b⁺)/(t_c-t_s);
30         if v_c > a_l+(t_c - t_s) + b⁺ + ε then // raise slope
31             a_l+ ← (v_c-ε-b⁺)/(t_c-t_s);
32     if diff > 0 then b_intervals[b⁻].add(⟨a_l-, a_u-, t_s⟩);
33     else b_intervals[b⁺].add(⟨a_l+, a_u+, t_s⟩);
34     return b_intervals;
```

---

a runtime overhead, which we manage via a parameter $q \in [0, 1]$ that limits candidate endpoints:

$$j_{\min} = \min \left\{ \arg\max_{j \in [i + r_i^q, i + r_i]} \{ j + r_{j+1} \} \right\} \tag{4}$$

The $r_i^q$ term in Equation (4) sets the minimum segment length. For $q = 0$, Equation (4) checks all eligible endpoints $j$, and, as our experiments show, gains space savings on par with or better than those of MINSEGMENTS. The growth of $q$ drops candidate endpoints near the segment's start. Intuitively, segments that underuse a starting point's reach are unlikely to serve in the shortest PLA sequence. By contrast, $q$ near 1 curbs both search space and compression. At $q = 1$, TAILORPIECEGD reduces to MIXPIECE, considering only the endpoint $i + r_i$ for a segment starting at $i$.

## 4 EXPERIMENTAL RESULTS

We ran experiments on a 3.3GHz Intel® Core™ i5-4590 machine with 6MB L3 cache and 16GB DDR3 1.6GHz RAM. We implemented[1] our algorithms in Java and compared performance against:

- Methods for the $L_\infty$ error metric:
  - SLIDE[2] (Elmeleegy et al., 2009), which optimally solves *disjoint PLA* using a convex hull.
  - MIXED [2] (Luo et al., 2015), which finds a least-length PLA of *mixed joint and disjoint* segments.
  - MIXPIECE[3] (Kitsios et al., 2024), the leading method for jointly representing PLA segments.

---

[1] https://anonymous.4open.science/r/pla-compression
[2] https://cse.hkust.edu.hk/~yike/PLAcode.rar
[3] https://github.com/xkitsios/Mix-Piece_Sim-Piece

- HIRE[4] (Barbarioli et al., 2023), which constructs a synopsis data structure through a recursion of partitioning, piecewise approximation, and residualization steps at increasingly finer granularity. HIRE, however, fails to meet the specified $L_\infty$ error threshold in 78% of the datasets in our experiments; therefore we evaluate it in terms of approximation quality by the $L_2$ error metric.

- Methods designed for the $L_2$ error metric:
  - Bottom-Up[5] (Keogh et al., 2001b), which merges in turn adjacent segments yielding least error.
  - PAA (Keogh et al., 2001a), which represent equi-sized segments, each with the mean of its values.
  - DFT (Cooley & Tukey, 1965), which uses the first few Discrete Fourier Transform features.
- Camel[6] (Yao et al., 2024), which separately compresses the integer and decimal parts of double-precision floating-point numbers, to a precision of four decimal places.

We evaluate solutions on all datasets of the UCR Time Series Classification Archive[7] that do not contain undefined values. Given that *lossless* algorithms (Yao et al., 2024) achieve compression ratios up to 4, we relegated data that cannot be compressed by a ratio of at least 10 with $\epsilon = 1\%$ as unsuitable for PLA-based compression. Used SLIDE (Elmeleegy et al., 2009) to ensure fairness, we compressed all data with a maximum error at $1\%$ of the signal's range and selected those that attained compression greater than 10, ending up with 41 datasets, which we use in our experiments. For completeness, we also report aggregate results for the entire archive.

## 4.1 PLA SEQUENCE LENGTH

We first assess PLA sequence lengths. Table 1 reports the number of produced segments as a percentage over the minimum disjoint PLA sequence length under a maximum error threshold (Luo et al., 2015) by SLIDE (Elmeleegy et al., 2009; O'Rourke, 1981), for ten values

Table 1: Segments over minimum disjoint PLA segments.

| $\epsilon$ | Slide (segments) | MIXPIECE | MINSEGMENTS | TAILORPIECEGD ($q = 0$) | TAILORPIECEDP ($p = 2^{-20}$) |
|---|---|---|---|---|---|
| 1% | 2786.0 | +9.3% | +6.6% | +6.7% | +6.6% |
| 2% | 1805.6 | +7.9% | +5.3% | +5.4% | +5.3% |
| 3% | 1419.0 | +7.9% | +5.5% | +5.6% | +5.5% |
| 4% | 1209.3 | +7.5% | +5.5% | +5.6% | +5.5% |
| 5% | 1051.1 | +8.3% | +6.0% | +6.1% | +6.0% |
| 6% | 939.7 | +8.2% | +6.1% | +6.2% | +6.1% |
| 7% | 839.2 | +8.9% | +6.7% | +6.8% | +6.7% |
| 8% | 758.4 | +8.4% | +6.6% | +6.7% | +6.6% |
| 9% | 696.9 | +9.1% | +7.4% | +7.5% | +7.4% |
| 10% | 646.3 | +8.8% | +7.0% | +7.2% | +7.0% |

of $\epsilon \in [1\%, 10\%]$. MIXPIECE produces PLA sequences 7.5–9.3% longer than the minimum, leaving room for improvement. MINSEGMENTS cuts this to 5.3–7.4%, minimizing disjoint segments with $\epsilon-$quantized starting points, which favor grouping, while SLIDE selects starting points freely, thus produces shorter PLA sequences. TAILORPIECEGD performs even better: despite its greedy strategy, it adds only 0.1–0.2% segments over MINSEGMENTS for $\epsilon = 1$–10%, yielding PLA lengths close to the optimum. TAILORPIECEDP produces the same segments as MINSEGMENTS for small $p$ (e.g., $2^{-20}$). Section 4.6 discusses the effect of $p$ in more detail.

## 4.2 COMPRESSION RATIO COMPARISON

Next, we compare our methods to error-bounded solutions SLIDE (Elmeleegy et al., 2009), MIXED (Luo et al., 2015) and MIXPIECE (Kitsios et al., 2024), and also report results for Camel (Yao et al., 2024) as a reference point for *lossless* representation. For brevity, we omit PMC-MR (Lazaridis & Mehrotra, 2003) and Swing (Elmeleegy et al., 2009), as they underperform MIXPIECE (Kitsios et al., 2024), and include Serf-XOR (Li et al., 2025), a *streaming* floating-point compressor with only modest savings. We measure *compression ratio* as the uncompressed representation size (values and timestamps, 8 bytes per point) over the compressed size.

Table 2 reports results for maximum error[8] 5% and 10% of the signal range for the 41 selected UCR archive time-series, with averages for the rest. Both MINSEGMENTS and TAILORPIECEGD improve compression ratio over MIXPIECE. On average, MINSEGMENTS yields 18% and 16% gains for 5% and 10% error, respectively, as MIXPIECE's myopic first phase hinders joint representation. TAILORPIECEGD attains comparable or greater savings, averaging 20% over MIXPIECE, by exploring a broader search space. More strikingly, TAILORPIECEGD surpasses MINSEGMENTS: although it creates more segments (Table 1), its larger slope intervals allow more joint representations.

---

[4] https://github.com/gmersy/HIRE

[5] https://github.com/NickFoubert/simple-segment

[6] https://github.com/yoyo185644/camel

[7] https://www.cs.ucr.edu/~eamonn/time_series_data/

[8] We omit detailed results for $\epsilon = 1\%$ due to space constraints, but present averages in Figure 11a.

Table 2: Compression ratio comparison for $\epsilon = 5\%$ and $\epsilon = 10\%$ of the signal's range.

| | CAMEL | 5% SLIDE | MIXED | MIXPIECE | MINSEGMENTS | TAILORPIECEGD ($q=0$) | TAILORPIECEDP ($p=2^{-20}$) | 10.0% SLIDE | MIXED | MIXPIECE | MINSEGMENTS | TAILORPIECEGD ($q=0$) | TAILORPIECEDP ($p=2^{-20}$) |
|---|---|---|---|---|---|---|---|---|---|---|---|---|---|
| Adiac | 3.83 | 24.65 | 36.00 | 53.44 | 58.24 | 58.81 | **63.25** | 29.02 | 43.41 | 75.00 | 80.61 | 80.23 | **83.98** |
| Beef | 3.95 | 41.20 | 53.41 | 66.75 | 77.50 | 77.42 | **85.91** | 70.06 | 88.54 | 113.65 | 133.57 | 135.25 | **147.93** |
| BirdChicken | 3.81 | 40.35 | 56.34 | 59.60 | 71.20 | 70.77 | **77.06** | 55.65 | 78.77 | 84.45 | 109.74 | 107.86 | **126.61** |
| Car | 3.82 | 55.00 | 78.64 | 89.00 | 108.83 | 109.80 | **125.27** | 85.43 | 122.66 | 146.54 | 178.00 | 176.63 | **214.53** |
| CinCECGTorso | 4.03 | 248.45 | 349.04 | 352.11 | 419.07 | 432.90 | **477.04** | 413.22 | 523.56 | 558.27 | 669.46 | 713.65 | **746.27** |
| DiatomSizeReduction | 3.82 | 46.87 | 69.42 | 93.60 | 110.10 | 108.14 | **122.19** | 56.72 | 85.07 | 135.96 | 152.41 | 151.00 | **164.54** |
| EthanolLevel | 3.46 | 152.44 | 224.22 | 224.09 | 282.88 | 286.02 | **348.43** | 233.65 | 327.33 | 332.36 | 421.50 | 437.16 | **528.40** |
| Fish | 3.81 | 92.42 | 115.61 | 168.31 | 203.15 | 214.65 | **222.10** | 308.17 | 312.01 | 414.29 | 440.77 | 463.77 | **517.46** |
| FreezerRegularTrain | 3.70 | 63.23 | 67.68 | 131.30 | 148.42 | 148.89 | **152.58** | 91.66 | 95.56 | 238.59 | 233.78 | 244.28 | **248.68** |
| Fungi | 3.65 | 33.32 | 47.87 | 64.98 | 69.39 | 72.54 | **80.50** | 47.93 | 67.78 | 104.74 | 102.00 | 107.73 | **121.49** |
| GunPoint | 3.70 | 22.93 | 33.22 | 44.73 | 52.08 | 50.97 | **58.20** | 33.71 | 44.38 | 57.69 | 65.22 | 68.95 | **74.01** |
| GunPointAgeSpan | 2.84 | 30.12 | 37.88 | 54.95 | 63.39 | 64.24 | **71.23** | 57.72 | 67.48 | 100.93 | 120.45 | 125.54 | **132.10** |
| GunPointMaleVersusFemale | 2.99 | 30.63 | 38.30 | 55.97 | 64.47 | 65.36 | **72.37** | 59.24 | 72.16 | 103.80 | 123.56 | 128.09 | **135.57** |
| GunPointOldVersusYoung | 2.85 | 31.42 | 39.09 | 56.40 | 65.28 | 65.44 | **73.03** | 60.43 | 69.38 | 104.18 | 127.40 | 129.72 | **138.17** |
| HandOutlines | 3.90 | 180.02 | 269.91 | 293.15 | 377.18 | 374.71 | **473.93** | 212.77 | 280.90 | 292.72 | 423.73 | 444.20 | **487.51** |
| Haptics | 3.82 | 105.76 | 122.32 | 171.90 | 206.61 | 210.53 | **223.84** | 247.53 | 307.69 | 402.62 | 478.18 | 481.93 | **510.86** |
| Herring | 3.77 | 41.60 | 59.17 | 69.53 | 85.33 | 85.47 | **96.29** | 66.57 | 90.08 | 117.21 | 145.19 | 146.00 | **166.02** |
| HouseTwenty | 4.71 | 23.07 | 24.62 | 65.37 | 65.12 | 65.52 | **65.61** | 29.13 | 29.87 | 76.42 | 76.58 | 77.23 | **78.34** |
| InlineSkate | 3.81 | 226.50 | 306.28 | 304.18 | 373.66 | 378.79 | **418.63** | 386.85 | 536.19 | 519.82 | 616.33 | 638.98 | **746.97** |
| LargeKitchenAppliances | 4.23 | 104.06 | 121.80 | 195.79 | 205.66 | 211.42 | **214.02** | 162.34 | 182.15 | 314.22 | 324.68 | 332.23 | **354.93** |
| Lightning2 | 3.90 | 80.00 | 100.17 | 141.95 | 154.08 | 159.54 | **169.40** | 185.95 | 226.03 | 313.00 | 330.27 | 358.71 | **373.48** |
| Mallat | 3.75 | 31.79 | 43.90 | 56.27 | 65.55 | 65.63 | **72.44** | 56.53 | 72.39 | 98.11 | 114.56 | 117.13 | **129.09** |
| Meat | 4.04 | 51.03 | 69.73 | 123.59 | 132.25 | 131.68 | **144.03** | 61.86 | 82.14 | 150.38 | 154.37 | 161.08 | **172.45** |
| MixedShapesRegularTrain | 3.84 | 67.32 | 94.30 | 104.41 | 123.80 | 124.15 | **143.88** | 95.60 | 128.95 | 149.25 | 188.19 | 188.28 | **221.42** |
| NonInvasiveFetalECGThorax1 | 3.79 | 78.59 | 116.35 | 142.73 | 169.28 | 176.91 | **194.46** | 114.09 | 150.26 | 209.81 | 239.81 | 258.98 | **286.33** |
| NonInvasiveFetalECGThorax2 | 3.79 | 75.99 | 113.19 | 139.25 | 163.03 | 171.27 | **184.97** | 115.08 | 146.31 | 209.15 | 235.71 | 253.49 | **278.45** |
| PigAirwayPressure | 3.93 | 199.20 | 287.36 | 296.52 | 390.24 | 391.39 | **425.99** | 284.09 | 416.67 | 442.97 | 569.80 | 643.09 | |
| PigArtPressure | 3.65 | 38.66 | 52.77 | 66.51 | 80.92 | 82.14 | **90.59** | 75.47 | 92.29 | 129.01 | 140.40 | 161.65 | **170.87** |
| PigCVP | 3.73 | 48.08 | 60.42 | 66.81 | 82.35 | 80.89 | **90.39** | 118.69 | 133.33 | 159.05 | 201.26 | 197.53 | **210.80** |
| Rock | 3.92 | 357.14 | 465.12 | 466.74 | 519.48 | 539.81 | **557.10** | 719.42 | 847.46 | 911.16 | 1012.66 | 1010.10 | **1084.01** |
| ShapesAll | 3.82 | 42.03 | 58.31 | 70.65 | 84.87 | 84.46 | **96.25** | 61.26 | 84.14 | 105.08 | 133.38 | 131.73 | **152.56** |
| SmallKitchenAppliances | 4.21 | 38.39 | 45.45 | 93.09 | 93.20 | 95.03 | **96.15** | 50.02 | 59.56 | 113.99 | 114.81 | 120.59 | **121.10** |
| StarLightCurves | 3.81 | 170.79 | 234.74 | 231.68 | 287.36 | 293.79 | **332.23** | 236.97 | 343.05 | 344.38 | 428.95 | 456.36 | **541.64** |
| Symbols | 3.81 | 53.19 | 70.95 | 86.68 | 102.59 | 104.78 | **117.23** | 79.52 | 106.44 | 145.83 | 176.80 | 179.01 | **198.91** |
| Trace | 3.69 | 49.46 | 67.20 | 104.31 | 112.65 | 117.96 | **123.77** | 75.86 | 93.86 | 166.41 | 168.78 | 183.49 | **190.48** |
| UMD | 4.58 | 17.61 | 25.45 | 42.16 | 41.27 | 44.19 | **45.37** | 31.51 | 34.62 | 66.69 | 69.72 | 73.87 | **76.49** |
| UWaveGestureLibraryX | 3.83 | 50.26 | 66.38 | 83.02 | 98.79 | 101.37 | **113.07** | 107.99 | 120.85 | 169.24 | 209.42 | 212.99 | **228.57** |
| UWaveGestureLibraryY | 3.80 | 45.53 | 59.47 | 76.23 | 88.17 | 90.49 | **99.96** | 88.73 | 107.58 | 141.94 | 177.31 | 179.13 | **192.91** |
| UWaveGestureLibraryZ | 3.84 | 38.68 | 51.15 | 65.80 | 76.20 | 78.05 | **86.26** | 71.05 | 87.18 | 114.91 | 141.37 | 144.01 | **157.26** |
| Wafer | 3.74 | 25.40 | 26.56 | 63.87 | 66.05 | 66.81 | **68.06** | 100.81 | 118.27 | 198.12 | 194.03 | 215.69 | **226.12** |
| Yoga | 3.82 | 37.15 | 51.83 | 63.88 | 76.00 | 77.43 | **86.59** | 51.91 | 71.12 | 95.82 | 116.01 | 116.03 | **134.00** |
| Average | 3.80 | 77.81 | 105.16 | 126.86 | 149.16 | 151.95 | **167.31** | 133.91 | 169.45 | 212.87 | 246.52 | 255.61 | **280.84** |
| Average (rest) | 3.79 | 13.54 | 17.50 | 28.70 | 30.95 | 31.35 | **32.99** | 29.70 | 35.60 | 58.24 | 63.22 | 64.99 | **68.19** |
| Average (all) | 3.79 | 38.40 | 51.41 | 66.67 | 76.67 | 78.00 | **84.94** | 70.00 | 87.37 | 118.05 | 134.12 | 138.72 | **150.44** |

Figure 6 shows the effect of using the *latest* instead of the *earliest* endpoint $j$ among those providing the largest combined reach in TAILORPIECEGD, replacing the min with max in Equation (4). This change nullifies TAILORPIECEGD's advantage over MINSEGMENTS, yielding segments less amenable to grouping. TAILORPIECEDP with small $p$ yields the best compression ratio on all datasets (including those not in Table 2), exceeding MIXPIECE by over 32% for $\epsilon$ at 5% and 10%, and reaching 34% with $\epsilon = 7\%$. Table 2 further shows that these gains hold over the entire UCR archive, even for datasets with high local variance. We omit results for the streaming method Serf-XOR (Li et al., 2025)[9] streaming compression method, as its average compression ratio—12.08 and 13.01 on our 41 time series, and 10.08 and 11.71 across all data, for 5% and 10% error, respectively—remains uncompetitive and does not improve significantly even with $\epsilon = 50\%$. We thus exclude Serf-XOR in the following.

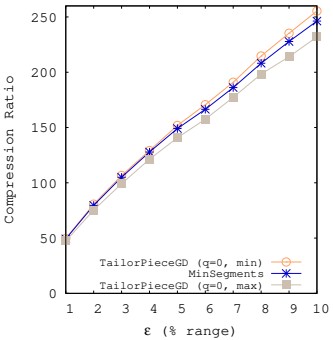

Figure 6: Using *max* instead of *min* in Equation (4) hurts performance.

[9] https://github.com/Spatio-Temporal-Lab/Serf

### 4.3 QUALITY OF APPROXIMATION

Our next experiment reports the Normalized Root Mean Squared Error (NRMSE) to assess approximation quality; we use normalized RMSE, as value ranges vary largely across datasets.

Figure 7 plots average NRMSE vs. compression ratio: Figure 7a for the selected datasets of Table 2 and Figure 7b for the full UCR archive. We include all algorithms with an $L_\infty$ threshold and three targeting $L_2$ (PAA (Keogh et al., 2001a), DFT (Cooley & Tukey, 1965), Bottom-Up (Keogh et al., 2001b)). For each compression ra-

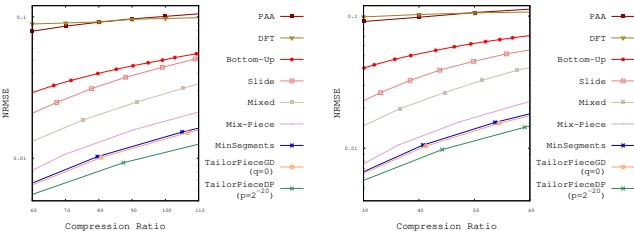

(a) Selected datasets of Table 2.    (b) All UCR archive datasets.

Figure 7: NRMSE vs. compression ratio; y-axis on log scale.

tio, algorithms with $L_\infty$ guarantees offer higher average quality than those targeting $L_2$. Our methods surpass the state of the art, achieving lower NRMSE than SLIDE, MIXED and MIXPIECE under the same space. TAILORPIECEGD slightly outperforms MINSEGMENTS, while TAILORPIECEDP delivers the best quality by a wide margin. Figure 8 visualizes TAILORPIECEDP segments for a Car dataset sample at $\epsilon = 1\%$ and 5%, illustrating how segment count and quality drop as $\epsilon$ rises.

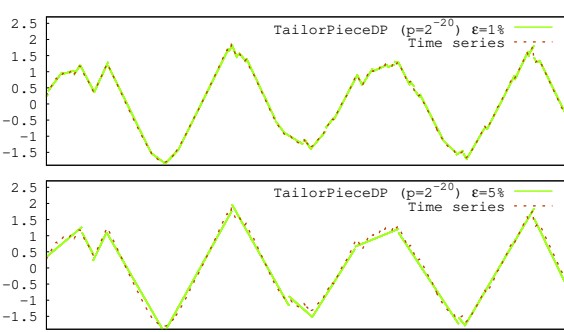

Figure 8: PLA segments, `Car` data sample.

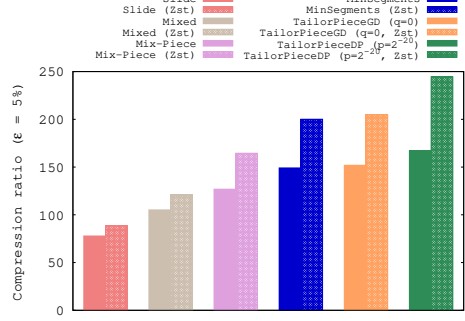

Figure 9: General compression on outputs.

### 4.4 GENERAL-PURPOSE COMPRESSION GAINS

Figure 9 shows the effect of lossless general-purpose compression, Zstandard (Collet, 2015), on outputs with $\epsilon = 5\%$. Our methods yield the largest overall savings. Figure 10 shows results for $L_2$-targeting algorithms—PAA (Keogh et al., 2001a), DFT (Cooley & Tukey, 1965), Bottom-Up (Keogh et al., 2001b), and HIRE (Barbarioli et al., 2023)—using ZStandard, except for HIRE, which uses TRC[10]. General-purpose compression boosts the quality-space tradeoff but retains the algorithm ranking: MINSEGMENTS, TAILORPIECEGD, and TAILORPIECEDP still offer the best tradeoff. HIRE performs significantly worse.

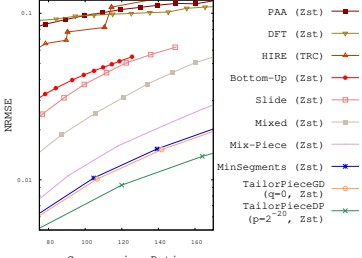

Figure 10: NMRSE with output compression; y-axis on log scale.

### 4.5 COMPRESSION/TIME TRADEOFF

Figure 11 reports compression times and ratios, averaged over the datasets of Table 2, with times normalized by dataset size. TAILORPIECEGD and TAILORPIECEDP traces show performance across $q \in [0, 0.99]$ and $p \in [0, 2^{-20}]$, with MINSEGMENTS corresponding to TAILORPIECEDP at $p = 0$. Our methods outperform competitors: TAILORPIECEDP at $p = 2^{-20}$ achieves the highest space savings, improving by 20%, 32% and 32% over MIXPIECE for $\epsilon = 1\%$, 5% and 10%. TAILORPIECEGD with $q = 0$ ranks second, improving by 13%, 20% and 20% over MIXPIECE. MINSEGMENTS (i.e., TAILORPIECEDP with $p = 0$) yields slightly worse compression than TAILORPIECEGD, despite shorter PLA sequences, due to larger slope intervals that favor grouping.

---

[10] https://github.com/powturbo/Turbo-Range-Coder

TAILORPIECEDP incurs a notable runtime overhead, as its compression time matches MINSEG-MENTS, making small-$p$ TAILORPIECEDP preferable when space is critical. TAILORPIECEGD with $q = 0$ runs two orders of magnitude faster than dynamic-programming approaches, offering a strong tradeoff between compression and runtime. Higher $q$ enhances this tradeoff, with compression time on par with MIXPIECE and competitive space savings. Thus, TAILORPIECEGD excels when speed matters. Figure 12 plots average decompression time, often more critical than compression time, as data is written once but read repeatedly. Our methods surpass MIXPIECE, while SLIDE and MIXED lag due to computing slope-intercept equations during decompression.

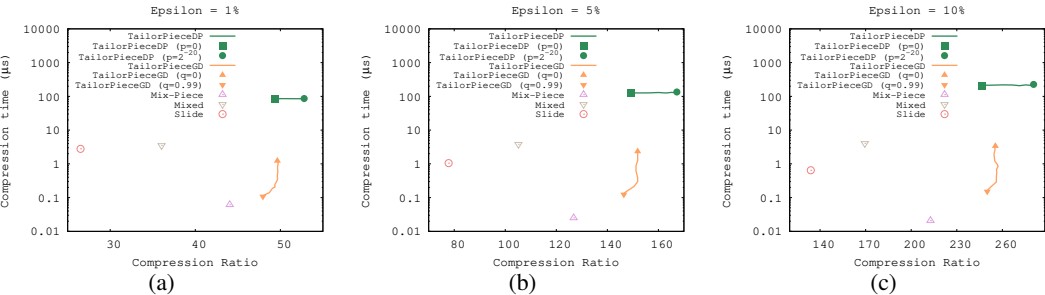

Figure 11: Compression time per data record vs ratio at different error thresholds.

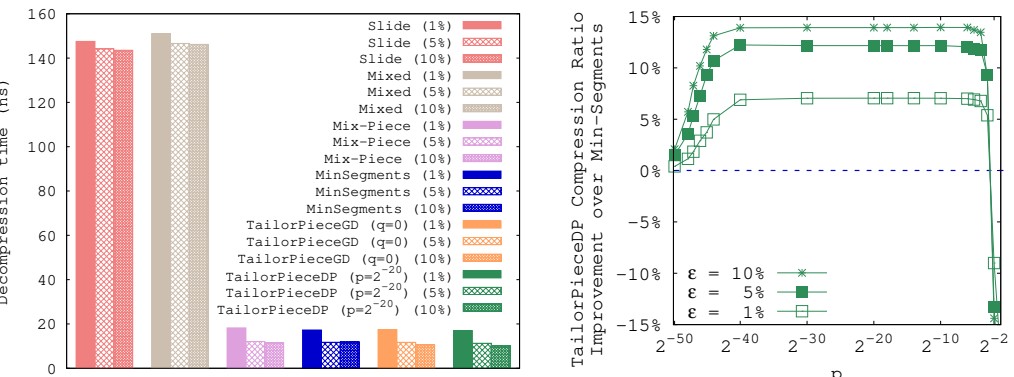

Figure 12: Decompression time per record.  Figure 13: Average improvement vs. $p$.

### 4.6 FAVORING SEGMENTS WITH LARGE INTERVALS

The dynamic-programming MINSEGMENTS algorithm minimizes PLA sequence length. However, compression also depends on slope intervals. Larger intervals increase the chance of grouping similar segments, boosting space savings, thus we favor segments with *large* slope intervals. Figure 13 (in the Appendix) illustrates the effect of exponent $p$ of TAILORPIECEDP, which adjusts the effect of slope interval size in the objective, on the datasets of Table 2. Large $p$ values hurt compression, as they drag TAILORPIECEDP to use more PLA segments (e.g., for $p = 2^{-1}$ and $\epsilon = 10\%$, the compression of TAILORPIECEDP is $15\%$ worse than MINSEGMENTS). However, smaller $p$ yields sequence length on par with MINSEGMENTS and also enlarges average interval size, as it is apparent for $p < 2^{-6}$, yielding space savings above $7\%$, $12\%$ and $13.9\%$, for $\epsilon$ equal to $1\%$, $5\%$ and $10\%$, respectively. The plateau beyond this point arises as $p$ is small enough for TAILORPIECEDP to match the optimal number of segments while creating larger slope intervals.

## 5 CONCLUSIONS

We presented techniques to build and compress piecewise linear segments for time-series storage: MINSEGMENTS computes a minimum-length PLA with quantized starting values under a maximum error threshold. TAILORPIECEDP refines this objective to render segments more amenable to joint representation by common starting values and overlapping slope intervals, yielding extra space savings. Drawing on these insights, TAILORPIECEGD offers a tunable tradeoff between compression and runtime by limiting its greedy-search space. Experiments show TAILORPIECEGD attains space savings near MINSEGMENTS, vastly exceeding the state of the art, and runs two orders of magnitude faster and on par with existing approaches.

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
