# OpenReview forum: "TailorPiece: Tailoring Linear Models for Joint Representation"
_ICLR.cc/2026/Conference — Submitted to ICLR 2026_

### Official Review · Reviewer_72nW · 2025-10-18

**Soundness:** 2
**Presentation:** 3
**Contribution:** 2
**Rating:** 2
**Confidence:** 3

**Summary:**

The paper addresses the problem of lossy time-series compression under a fixed error bound with quantized starting values. It identifies limitations in the previous MIXPIECE method, which greedily maximizes segment length but may yield suboptimal segment counts and poor mergeability. They propose three variants: MINSEGMENTS, a dynamic programming approach for globally minimal segmentation under quantization; TAILORPIECEDP, which further optimizes for mergeability by maximizing slope interval width; and TAILORPIECEGD, a greedy lookahead version balancing compression quality and runtime.

**Strengths:**

- The paper clearly articulates the practical need to improve MIXPIECE’s segmentation under quantization and mergeability constraints.

- The DP formulations are logically consistent and correctly defined for global optimality. The overall structure and explanations are straightforward, making the method easy to reproduce and understand.

- The approach provides a tunable trade-off between accuracy and efficiency, which is valuable for real-world compression systems.

**Weaknesses:**

- While technically correct, the paper’s contributions are incremental extensions of existing piecewise linear approximation (PLA) frameworks rather than fundamentally new ideas, which limits its novelty. No new theoretical model, loss formulation, or probabilistic insight into PLA is introduced. The paper essentially re-optimizes an existing heuristic with better parameterization. From a research standpoint, this positions the contribution as an engineering refinement, not a methodological breakthrough.

- The paper offers no formal complexity analysis, approximation guarantees, or theoretical characterization of how the new objectives affect global optimality. For example, while MINSEGMENTS claims “globally minimal” segmentation, the proof is implicit. There is no formal definition of optimality under quantized constraints or derivation of time/space complexity. Similarly, for TAILORPIECEDP, the trade-off between segment count and interval width is handled empirically but never quantified analytically.

- The evaluation focuses on synthetic or benchmark datasets (UCR) with standard metrics and lacks deeper diagnostic analysis.

- No discussion on integration into full compression or streaming pipelines.

**Questions:**

- How does the algorithm scale with very long or streaming sequences—can the DP version handle data in the order of millions of points?

- Can the authors provide theoretical or empirical bounds linking segment count and slope interval width?

- How sensitive is performance to quantization granularity or dataset smoothness?

- Could the approach extend to multidimensional or irregularly sampled time series?

---

> ### Author Response · Authors · 2025-11-15
> **The algorithms are methodologically novel; not existing heuristics**
>
> Thank you for the thorough and constructive review.
>
> >The paper essentially re-optimizes an existing heuristic with better parameterization.
>
> The algorithms we introduce are methodologically novel; they are not existing heuristics.
>
> >no formal complexity analysis, approximation guarantees
>
> We offer the time complexity in Lines 195-196. Since MinSegments and TailorPieceDP compute the optimal values of their objectives, no approximation guarantees are applicable.
>
> >while MINSEGMENTS claims “globally minimal” segmentation, the proof is implicit.
>
> The proof following by induction. We have now added it in the uploaded revised version: By the inductive hypothesis, for all $j > i$, $L(j)$ gives the minimum PLA length for the subsequence starting at $t_j$. Each valid endpoint $j$ of the first segment, $i < j \le i + r_i$, yields $L(j+1) + 1$ segments in total. Minimizing over all such $j$ therefore gives the minimum PLA length starting from $t_i$.
>
> >no formal definition of optimality under quantized constraints
>
> Optimality under quantized constraints refers to achieving the best possible value of the objective when the segment starting points are restricted to a discrete (quantized) set of positions. It is the standard notion of optimality applied within the additional constraint that segment starts must lie on the quantized grid.
>
> >the trade-off between segment count and interval width is never quantified analytically.
> >provide theoretical or empirical bounds linking segment count and slope interval width?
>
> This tradeoff arises because, for a segment to satisfy the error threshold, its permissible slope interval is a non-increasing function of segment length: longer segments can only shrink the allowed angle range. Increasing the angle interval to allow more flexibility typically requires shortening segments, which increases the segment count. Thus, segment count and angle interval width are inherently inversely related.
>
> >No discussion on integration into full compression or streaming pipelines.
>
> A compression or streaming system builds on top of a PLA stage by encoding segment parameters, applying quantization, and performing entropy coding or incremental streaming; our contribution focuses solely on the PLA stage.
>
> >can the DP version handle data in the order of millions of points?
>
> The DP version has time complexity linear in $n$. As Figure 11 shows, TailorPieceDP needs 100 μs per data item, hence a data set of 1 million points requires 100 s.
>
> >How sensitive is performance to quantization granularity?
>
> We always apply quantization by the error threshold ε; as our results with different values of ε show, performance is robust with respect to quantization granularity.
>
> >Could the approach extend to multidimensional or irregularly sampled time series?
>
> Yes, our techniques could extend to multidimensional and irregularly sampled data. For multidimensional series, segments could be defined either per dimension or using a multidimensional error metric. For irregularly sampled series, segment slopes would consider the actual time intervals. Exploring these extensions is an interesting direction for future work.

---

### Official Review · Reviewer_RHoX · 2025-10-28

**Soundness:** 3
**Presentation:** 2
**Contribution:** 1
**Rating:** 2
**Confidence:** 4

**Summary:**

The paper proposes a library of algorithms for lossy-PLA compression of time-stamped data. The aim of the work is to reduce the storage requirement, and thereby, the cost, by achieving better compression rates through improving the joint representation of similar data. The algorithms were tested against 8 baselines using 41 datasets. The experiments demonstrated improvements in compression rate in addition to runtime improvements for some algorithms.

**Strengths:**

- The paper attempts to address an important research area regarding storing big time-stamped data with the lowest possible storage cost, which is applicable to many sectors.
- The experiments were conducted against 8 baselines.
- The results show improvements compared with the baselines.
- The algorithms presented in the paper are well-explained both in text and mathematically.

**Weaknesses:**

- Very brief presentation and discussion of related work. The paper lacks a proper “related work” section, where the work from the literature is typically presented, discussed, and research gaps are listed. Without such section, it is not clear where the contribution of this paper stands in relation to the related literature, and whether it addresses an actual gap or not. There are scattered information about related methods across the paper; however, they do not properly replace a proper “related work” section.
- The contribution of the paper is not clear. No specific research questions, clear setup of experiments, nor experiments goals are clearly presented.
- When comparing the algorithms presented in the paper with the baselines, the results are presented in numerical forms without testing whether the improvements are statistically significant or not. The statistical significance of the claimed improvements in comparison with the baseline methods needs to be tested (using a Friedman test followed by Nemenyi post-hoc test, for example).
Relatively weak benchmark. The experiments were run on a subset of datasets from the UCR Archives (41 out of 128 datasets). The authors mention that only datasets that do not contain undefined values were used. This is an issue for two reasons: (i) this is not always the case for real-world data, especially the sectors mentioned by the authors in the introduction (line 033); it is better to test the algorithms using all the datasets in the UCR Archive by passing the data as it is or preprocessing the data to handle the undefined values, or both. (ii) - The authors of the UCR Archive explicitly discourage against cherry-picking datasets from the archive [1].
- The implications of lossy compression are not properly discussed. The paper proposed lossy compression as the solution for storing time-stamped data in a more efficient way compared with lossless compression. However, the implications of such choice are not properly discussed. For example, how accurate the original data can be reconstructed from the compressed representation and how severe do the lost information affect the usability of the data in any downstream tasks. In this end, it is important to be able to use the data after storing it.
- Wording issues in the text that lead to ambiguity:
Lines 061–062: “TAILORPIECEDP, which, building on top of TAILORPIECEDP,”

- Note: I understand that some sections might have been omitted due to the page limit. However, the authors could have made a better use of the appendix regarding the distribution of the content instead of omitting crucial details.

[1] Dau HA, Bagnall A, Kamgar K, Yeh CC, Zhu Y, Gharghabi S, Ratanamahatana CA, Keogh E. The UCR time series archive. IEEE/CAA Journal of Automatica Sinica. 2019 Nov 8;6(6):1293-305

**Questions:**

- Why is the complexity of the time series data sample not considered when attempting to extract compressed representations? It might be beneficial to consider the complexity of the data when calculating the minimum-length PLA in Algorithms 3.2, for example (See, e.g. [2])
- In all the algorithms presented in the paper, there is an error threshold (ε) as an input. However, it is not clear from the paper nor the appendix how this error threshold is calculated in the experiments and how can any future users of the algorithms calculate the error threshold. There is one brief explanation about this in lines 315–317, but it is still not clear how the values of the range are selected. Can you please elaborate on this? (the authors are encouraged to add a section about this in the paper or the appendix in any future versions).

[2] Nagaraj, N., Balasubramanian, K. & Dey, S. A new complexity measure for time series analysis and classification. Eur. Phys. J. Spec. Top. 222, 847–860 (2013). https://doi.org/10.1140/epjst/e2013-01888-9

---

> ### Author Response · Authors · 2025-11-15
> **Table 2 shows gains over the entire UCR archive**
>
> Thank you for the thorough and constructive review.
>
> >The paper lacks a proper “related work” section
>
> In our uploaded revision, we have expanded Section 2 to a more extensive related work section given the additional page.
>
> >where the contribution of this paper stands in relation to the related literature
>
> The contribution of this paper stands in the points listed in the introduction:
>
> 1. The first algorithm that returns a minimum-length PLA representation with quantized starting values under an error threshold.
>
> 2. An algorithm that produces segments with wide permissible slope intervals to enhance their mergeability, and merges them. Such algorithm has not been proposed in prior work.
>
> 3. An algorithm that selects each segment’s end to maximize the next segment’s reach and allows tuning its endpoint search space to trade runtime for compression. Such algorithm has not been proposed in prior work.
>
> Our algorithms improve the average compression ratio by up to 34% over the state of the art.
>
> >testing whether the improvements are statistically significant.
>
> As TailorPieceDP ranks first on every dataset (Table 2), the Friedman test will report a highly significant overall difference, and a Nemenyi post-hoc comparison will, by construction, declare TailorPieceDP significantly superior to all remaining methods. In such a deterministic ranking pattern, these tests add no substantive insight beyond what is already evident from the table. TailorPieceDP consistently dominates the alternatives, and the formal tests do not alter or refine the interpretation.
>
> >test the algorithms using all the datasets in the UCR Archive
>
> Table 2 shows that the gains hold over the entire UCR archive, even for datasets with high local variance and missing values.
>
> >how accurately the original data can be reconstructed from the compressed representation
>
> All our experiments report the error threshold, which reflects how accurately the original data can be reconstructed from the compressed representation.
>
> >Lines 061–062: “TAILORPIECEDP, which, building on top of TAILORPIECEDP,”
>
> We regret this typo. Please read: "on top of MinSegments".
>
> >consider the complexity of the data when calculating the minimum-length PLA
>
> We are unsure how ‘data complexity’ would factor into computing a minimum-length PLA. The number of segments is determined by the signal and the error bound; complexity measures may describe a time series but cannot affect the optimal PLA under an error constraint. Since our work concerns compression rather than analysis, it is unclear how such a measure would enter the formulation. If the reviewer has a specific formalization in mind, we would be glad to consider it.
>
> >how this error threshold is calculated in the experiments
>
> The error threshold is simply set by the user; we set it as a percentage of the signal's value range.
>
> >it is still not clear how the values of the range are selected.
>
> We chose values from 1% to 10% with a step of 1% to cover a wide range of possible requirements. Error threshold values below 1% lead to compression ratios on par with lossless schemes, while values above 10% lead to high inaccuracy.

---

### Official Review · Reviewer_6z6d · 2025-10-31

**Soundness:** 2
**Presentation:** 2
**Contribution:** 2
**Rating:** 2
**Confidence:** 4

**Summary:**

This paper proposes TAILORPIECE, a method that performs piecewise linear approximation (PLA) under a maximum error constraint, while balancing the minimal number of segments and segment mergeability. By combining dynamic programming and greedy strategies, the approach effectively improves the compression ratio while maintaining approximation accuracy.

**Strengths:**

S1. The framework explicitly optimizes for segment mergeability, allowing similar line segments to be grouped and jointly represented, thereby improving compression efficiency.

S2. Two hyperparameters (p and q) are introduced to flexibly balance compression accuracy and runtime, enabling users to tune the method
according to application needs.|

S3. TailorPiece advances beyond the previous state-of-the-art O(nlogn) complexity by introducing algorithms with O(Rn).

**Weaknesses:**

W1. The TAILORPIECEGD algorithm is a heuristic greedy approach that does not guarantee global optimality. Moreover, its performance depends on two hyperparameters p and q, which require manual tuning to achieve the desired trade-off between compression accuracy and runtime.

W2. TailorPiece quantizes the segment starting value $v$ into discrete levels defined by the error bound $\varepsilon$, using
$$
b^- = \lfloor v / \varepsilon \rfloor \times \varepsilon, \quad
b^+ = \lceil v / \varepsilon \rceil \times \varepsilon.
$$
when $v$ is an exact multiple of $\varepsilon$, the lower and upper bounds collapse ($b^- = b^+$), leaving no feasible interval for numerical tolerance.

W3. The baseline HIRE (Barbarioli et al., 2023) explicitly defines its reconstruction constraint in terms of L-infinity rather than L2. Therefore, classifying HIRE under L2-based methods could be misleading.

**Questions:**

Q1. The compression ratios of some datasets in Table 2 (such as Rock and CinCECGTorso) are very high. Is this related to the data characteristics?

---

> ### Author Response · Authors · 2025-11-15
> **TailorPieceGD controls the tradeoff with one parameter only**
>
> Thank you for the thorough and constructive review.
>
> >W1. TAILORPIECEGD ... depends on two hyperparameters p and q, which require manual tuning to achieve the desired trade-off.
>
> TailorPieceGD uses parameter $q$ only, to control the tradeoff.
>
> >W2. when $v$ is an exact multiple of $\epsilon$, the lower and upper bounds collapse, leaving no feasible interval.
>
> These are simply quantized values, not bounds. A single quantized value suffices to construct an approximation.
>
> >W3. HIRE (Barbarioli et al., 2023) explicitly defines its reconstruction constraint in terms of L-infinity rather than L2.
>
> Barbarioli et al. [Section 5.3.1] specify that they focus on L1 and L2 norms.
>
> >Q1. The compression ratios of some datasets in Table 2 are very high. Is this related to the data characteristics?
>
> Yes, the compression ratio depends on the data. Data in piecewise-linear form are more amenable to PLA-based approximation.

---

> > ### Comment · Reviewer_6z6d · 2025-11-27
> > **Response to Authors's Rebuttal**
> >
> > Thanks for the response. While Barbarioli et al. [Section 5.3.1]  mention that the approach can be generalized to L1 and L2 norms, HIRE' s core contribution remains explicitly anchored to the L-infinity criterion. The discussion of L1/L2 is a theoretical extension rather than the primary focus. Accordingly, it may be more appropriate not to place HIRE in the same category as baselines that are optimized exclusively for L2 error.

---

> ### Author Response · Authors · 2025-11-27
> **HIRE does not meet the error threshold it is supposed to enforce**
>
> Thank you for this observation. Indeed, when Barbarioli et al. write that they "focus on $L_1$ and $L_2$ norms", they mean that they focus on those norms among norms other than $L_\inf$ in the context of Section 5.3.1.
>
> Nevertheless, we had excluded HIRE as an $L_\inf$-oriented method as it does not, in practice, meet the error threshold it is supposed to enforce. The representations it produces fail to meet the specified $L_\inf$ error threshold $\epsilon$ in 78% of the datasets in our experiments. Given this predicament, it would be misleading to report HIRE's compression ratio under a supposedly achieved $L_\inf$ threshold. By contrast, it remains meaningful to report results for HIRE on the RMSE it attains as a function of compression ratio, which is what we present in Figure 10.
>
> We now listed HIRE among the methods focusing on $L_\inf$ and explained why we evaluate it in terms of approximation quality by the $L_2$ error metric.
>
> Thank you again for asking for further clarity on this matter.

---

### Official Review · Reviewer_DXa6 · 2025-10-31

**Soundness:** 3
**Presentation:** 3
**Contribution:** 3
**Rating:** 6
**Confidence:** 3

**Summary:**

The paper describes a suite of techniques for the Piecewise linear approximation.
The techniques are well described with definitions and problems.
The performances are compared with MIXpiece algorithm for the same task. Multiple experiments on compression and approximation are presented.

The proposed techniques are a slight variation on similar optimization
For example, in Figure 11, the TailorPieceGD shows a very similar compression ratio of TailorPieceDP with a reduced compression time. The best compression of TailorPieceDP is better but the improvement is less than 10%.
The comparisons with other techniques than MiXPiece are limited. Also other techniques base on euristics could be used for the same problem.

**Strengths:**

The approximation with sequence of values with Piecewise linear approximation achieves good results and the proposed techiques are better than MIxPiece technique

**Weaknesses:**

The comparison with other techniques also from other optimization paradigms is limited

**Questions:**

Can the algorithm of the proposed suite be integrated into a single technique that is optimized adaptively according to the approximation error?

---

> ### Author Response · Authors · 2025-11-15
> **Figure 10 presents a comparison to the current state-of-the-art compression methods**
>
> Thank you for the thorough and constructive review.
>
> >The comparison with other techniques than MiXPiece is limited.
>
> As MixPiece has been established as the SotA in PLA-based approximation, the comparison to MixPiece subsumes comparison to prior methods. Still, Figure 10 presents a comparison to the current state-of-the-art compression methods, including HIRE.
>
> >Can the algorithm be integrated into a single technique that is optimized adaptively according to the approximation error?
>
> Yes, the algorithms already accommodate and adapt to a target approximation error.

---

### Meta-Review · Area_Chair_QT7N · 2026-01-08

**Summary:**

This paper introduces TailorPiece, a set of algorithms for lossy piecewise linear approximation that tailor segment construction for joint representation under quantization constraints. Reviewers acknowledge that the paper is clearly written, technically sound, and demonstrates empirical compression gains over prior PLA methods. However, the majority of reviewers raise significant concerns about limited novelty, viewing the contributions as incremental refinements or re-optimizations of existing heuristics rather than fundamentally new methodology. Additional weaknesses include insufficient theoretical analysis, debated baseline choices, and limited experimental depth. While the authors provided detailed rebuttals and clarifications, these responses do not fully address the core concerns regarding contribution and impact. Therefore, the final recommendation is reject.

**Reviewer Concerns:**

Addressed by the rebuttal:
Several technical clarifications were adequately addressed. In particular, the authors clarified misunderstandings about the novelty of the proposed algorithms versus existing heuristics, added a formal proof for the MINSEGMENTS optimality claim, and expanded the related work section to better position the contribution. Concerns about the treatment of the HIRE baseline and error metrics were also reasonably explained.

Still outstanding:
Key concerns about limited novelty remain, as multiple reviewers still view the work as an incremental or engineering-oriented improvement rather than a conceptual advance. In addition, the lack of deeper theoretical analysis (e.g., formal characterization of trade-offs) and limited discussion of broader applicability and impact (e.g., streaming systems, downstream tasks) were not fully resolved by the rebuttal.

**Reviewer Scores:**

Reviewer DXa6: Likely would have kept the score roughly the same. The rebuttal addressed the comparison concerns, but the reviewer already viewed the gains as modest and novelty as limited, so a significant upward revision is unlikely.

Reviewer 6z6d: Unlikely to change their reject score. Although some technical points were clarified, the reviewer maintained concerns about optimality, baseline categorization (e.g., HIRE), and methodological strength even after follow-up discussion.

Reviewer RHoX: Likely would maintain a reject score. While the authors expanded the related work and clarified experimental choices, the reviewer’s main concerns about unclear contribution, experimental rigor, and missing significance analysis remain largely unresolved.

Reviewer 72nW: Unlikely to increase the reject score. Despite added proofs and clarifications, the reviewer’s core position—that the work represents an incremental engineering refinement with limited research novelty—would probably remain unchanged.

---

### Decision · Program_Chairs · 2026-01-26

Reject